# Sidedness-Dependent Prognostic Impact of Gene Alterations in Metastatic Colorectal Cancer in the Nationwide Cancer Genome Screening Project in Japan (SCRUM-Japan GI-SCREEN)

**DOI:** 10.3390/cancers15215172

**Published:** 2023-10-27

**Authors:** Takeshi Kajiwara, Tomohiro Nishina, Riu Yamashita, Yoshiaki Nakamura, Manabu Shiozawa, Satoshi Yuki, Hiroya Taniguchi, Hiroki Hara, Takashi Ohta, Taito Esaki, Eiji Shinozaki, Atsuo Takashima, Yoshiyuki Yamamoto, Kentaro Yamazaki, Takayuki Yoshino, Ichinosuke Hyodo

**Affiliations:** 1Department of Gastrointestinal Medical Oncology, National Hospital Organization Shikoku Cancer Center, Matsuyama 791-0280, Japan; nishina.tomohiro.nj@mail.hosp.go.jp (T.N.); ihyodo0@gmail.com (I.H.); 2Division of Translational Informatics, Exploratory Oncology Research and Clinical Trial Center, National Cancer Center, Kashiwa 277-8577, Japan; riuyamas@east.ncc.go.jp; 3Department of Gastroenterology and Gastrointestinal Oncology, National Cancer Center Hospital East, Kashiwa 277-8577, Japan; yoshinak@east.ncc.go.jp (Y.N.); tyoshino@east.ncc.go.jp (T.Y.); 4Department of Gastrointestinal Surgery, Kanagawa Cancer Center, Yokohama 241-8515, Japan; shiozawam@kcch.jp; 5Department of Gastroenterology and Hepatology, Hokkaido University Hospital, Sapporo 060-8638, Japan; satoshi-yuuki175@joy.ocn.ne.jp; 6Department of Clinical Oncology, Aichi Cancer Center Hospital, Nagoya 464-8681, Japan; hiroya.taniguchi@aichi-cc.jp; 7Department of Gastroenterology, Saitama Cancer Center, Kitaadachi-gun, Saitama 362-0806, Japan; hirhara@saitama-pho.jp; 8Department of Clinical Oncology, Kansai Rosai Hospital, Amagasaki 660-8511, Japan; tks.ohta@gmail.com; 9Department of Gastrointestinal and Medical Oncology, National Hospital Organization Kyushu Cancer Center, Fukuoka 811-1395, Japan; esaki.taito.fz@mail.hosp.go.jp; 10Department of Gastroenterological Chemotherapy, Cancer Institute Hospital of Japanese Foundation for Cancer Research, Tokyo 135-8550, Japan; eiji.shinozaki@gmail.com; 11Department of Gastrointestinal Medical Oncology, National Cancer Center Hospital, Tokyo 104-0045, Japan; atakashi@ncc.go.jp; 12Department of Gastroenterology, Faculty of Medicine, University of Tsukuba, Tsukuba 305-8575, Japan; yamamoto.yoshiyuki0621@gmail.com; 13Division of Gastrointestinal Oncology, Shizuoka Cancer Center, Shunto-gun, Shizuoka 411-8777, Japan; k.yamazaki1121@gmail.com

**Keywords:** *TP53*, gain-of-function variant, *NOTCH3*, next-generation sequencing, prognostic factor, colorectal cancer

## Abstract

**Simple Summary:**

The treatment strategies and prognoses of patients with metastatic colorectal cancer differ according to the sidedness of the primary tumor. The aim of this study was to evaluate the sidedness-dependent prognostic impact of gene alterations in metastatic colorectal cancer. Among patients diagnosed with metastatic colorectal cancer enrolled from April 2017 to March 2019, 531 patients who underwent complete gene sequencing were assessed. *TP53* gain-of-function and *KRAS* variants were poor prognostic factors, while the *NOTCH3* sole variant was a favorable prognostic factor for left-sided metastatic colorectal cancer. The *TP53* non-gain-of-function variant, *BRAF* V600E, and *MYC* amplification were poor prognostic factors for right-sided metastatic colon cancer. Prognostic gene alterations in metastatic colorectal cancer differed according to the sidedness of the primary tumor.

**Abstract:**

The treatment strategies and prognoses of patients with metastatic colorectal cancer (CRC) differ according to the sidedness of the primary tumor. *TP53* gain-of-function (GOF) and non-GOF variants have been reported to be differentially associated with prognosis by sidedness. We aimed to evaluate the sidedness-dependent prognostic impact of gene alterations in metastatic CRC. Patients enrolled between April 2017 and March 2019 were included in this study. Those excluded were individuals whose tumor tissues were obtained after chemotherapy and those who were enrolled in the study more than six months after starting first-line chemotherapy. Finally, we assessed 531 patients who underwent complete gene sequencing. The study revealed a significant difference in overall survival between individuals with left-sided CRC (*n* = 355) and right-sided colon cancer (CC) (*n* = 176) when considering the *TP53* non-GOF variant, *KRAS* wild-type, *NOTCH1* wild-type, *NOTCH1* covariant, *NOTCH3* sole variant, and *MYC* amplification. Multivariate analysis on each side revealed that the *TP53* GOF and *KRAS* variants were independent poor prognostic factors for left-sided CRC (*p* = 0.03 and *p* < 0.01, respectively), and the *TP53* non-GOF variant, *BRAF* V600E, and *MYC* amplification for right-sided CC (*p* < 0.05, *p* < 0.01, and *p* = 0.02, respectively). The *NOTCH3* sole variant was an independent and favorable prognostic factor for left-sided CRC (*p* < 0.01). The prognostic significance of gene alterations differed between left-sided CRC and right-sided CC.

## 1. Introduction

Left- and right-sided colorectal cancers (CRCs) differ in their underlying clinicopathological and biological features, such as age, sex, histology, gene variants (*RAS*, *BRAF*, *PIK3CA*, and *TP53*), microsatellite instability, chromosomal instability, gene methylation, and consensus molecular subtypes [1,2,3,4]. The sidedness of CRC likely serves as a surrogate marker for these features and prognostic factors in CRC. Patients with right-sided colon cancer (CC) have poorer prognoses than those with left-sided CRC, according to findings from an extensive meta-analysis and several randomized clinical trials [5,6,7,8]. *RAS* wild-type patients with left-sided CRC can share survival benefits from anti-epidermal growth factor receptor (EGFR) antibody therapy, whereas those with right-sided CC cannot [9]. Although these patients make up less than 10% of the total, those with *BRAF* V600E variant and microsatellite instability-high as well as deficient mismatch repair (common in right-sided CC) can benefit from BRAF inhibitors combined with anti-EGFR antibodies and immune checkpoint inhibitors, respectively. Similarly, patients with human epidermal growth factor receptor 2 amplification (more common in left-sided CRC) can also benefit from its blockade [10]. Thus, the sidedness of CRC implies the prognosis and efficacy of biological therapy.

Previous studies have shown that *TP53* gain-of-function (GOF) variants confer increased cell invasion, proliferation, chemoresistance, colony formation, genomic instability, and angiogenesis in various tumors [11,12,13,14]. *TP53* GOF variants have been suggested to extensively lose their original transcriptional function and gain functions, such as oncogenes, whereas *TP53* non-GOF variants preserve a certain level of the original transcriptional function. The prognostic significance of *TP53* GOF and non-GOF variants by the sidedness of metastatic CRC (mCRC) has recently been investigated [15]. *TP53* non-GOF variants were associated with poorer prognosis in right-sided CC versus left-sided CRC, while *TP53* GOF versus non-GOF variants were associated with poorer prognosis in left-sided CRC, but not in right-sided CC. However, these findings have not been confirmed by other studies.

To perform precision medicine for patients with advanced gastrointestinal cancers, we launched a Nationwide Cancer Genome Screening Project in Japan (SCRUM-Japan GI-SCREEN) that aims to enroll patients in suitably matched clinical trials [16]. SCRUM-Japan GI-SCREEN has been utilized to accelerate clinical trials and for novel research related to cancer biology. The genomic landscape and clinical outcomes according to the primary site in mCRC have been extensively studied and reported [17]. The *TP53* variant was more commonly observed in left-sided CRC, whereas the *KRAS* and *BRAF* variants were more prevalent in right-sided CC. In this study, the *NOTCH3* sole variant without the covariant of *NOTCH1* or *NOTCH2* is an independent favorable prognostic factor [18].

Treatment strategies and prognoses in patients with mCRC differ in terms of sidedness. However, how gene alterations, including *TP53* GOF/non-GOF and *NOTCH3* variants, affect the prognosis in each side remains unclear. In this study, we evaluated the sidedness-dependent prognostic impact of gene alterations in patients with mCRC.

## 2. Materials and Methods

### 2.1. Study Design and Patients

This was an observational, retrospective, multicenter study involving patients with mCRC. In total, 1777 patients with mCRC were enrolled in the SCRUM-Japan GI-SCREEN trial. The primary eligibility criteria were as follows: confirmed diagnosis of colorectal adenocarcinoma through pathology; an Eastern Cooperative Oncology Group performance status of 0–1; adequate bone marrow, renal, and hepatic function; *RAS* mutational status identified by polymerase chain reaction (PCR); planned or received systemic chemotherapy for metastatic disease; and written informed consent provided for participation in this study. Right-sided CC was defined as cancer of the ascending and transverse colon, and left-sided CRC was defined as cancer from the splenic flexure to the rectum. The target gene sequences of tumor tissue samples were determined using next-generation sequencing. The ethical, medical, and scientific aspects of the study were reviewed and approved by the institutional review board of each institution. The trial was registered in the University Hospital Medical Information Network Clinical Trials Registry (UMIN000016343). This study was conducted in accordance with the Declaration of Helsinki, as revised in 2000.

### 2.2. Targeted Sequencing

Formalin-fixed and paraffin-embedded biopsy or surgically resected samples were sent to the Clinical Laboratory Improvement Amendments-certified Life Technologies Clinical Services Laboratory (910 Riverside Parkway, West Sacramento, CA 95605, USA). Tumor DNA and RNA were extracted and subjected to multiplex PCR-based amplicon sequencing using the Ion Torrent™ Oncomine™ Comprehensive Assay v3 (Thermo Fisher Scientific, Waltham, MA, USA). This assay covers 161 of the most relevant cancer-related genes and detects their relevant single-nucleotide variants, copy number variations, gene fusions, and indels using one streamlined workflow (Appendix A). A gene variant was called when its allele frequency exceeded 5% and its coverage depth was above 200 reads, excluding synonymous variants. Gene amplification was considered to have occurred when a gene copy number was ≥4.0. Driver mutation classifications, such as GOF or loss-of-function, were determined using the Oncomine Knowledgebase and annotated using Ion Reporter™ software. The annotated genome variant call format files and the binary versions of the sequence alignment files were stored at the SCRUM-Japan Data Center.

We evaluated potential driver genes in receptor tyrosine kinase, RAS-MAPK, PI3K-AKT, TP53, WNT, TGF-β, and DNA damage repair signaling pathways. For *NOTCH1*, *NOTCH2*, and *NOTCH3*, variants of unknown significance were included. We defined the *TP53* R175H, R248W, R248Q, R249S, R273H, R273L, and R282W variants as GOF and other *TP53* variants as non-GOF in our dataset based on previous literature [15]. *NOTCH1*, *NOTCH2*, and *NOTCH3* variants were classified as sole variants or covariants with other *NOTCH* variants, based on our previous study [18].

### 2.3. Statistical Analyses

The significance of age and copy number differences was estimated using the Mann–Whitney U test. Differences in proportions were evaluated using the two-sided Fisher’s exact test. Overall survival (OS) was defined as the time from the initiation of first-line chemotherapy until death from any cause. Survivors were censored at their final contact. OS was calculated using the Kaplan–Meier method and compared using a log-rank test. Hazard ratios (HRs) and 95% confidence intervals (CIs) were estimated using a Cox proportional hazards model. The prognostic factors were evaluated using multivariate analyses using the Cox proportional hazards model. All statistical analyses were performed using the EZR version 1.61 [19]. Two-sided tests were used to calculate all *p*-values, and statistical significance was considered at *p* < 0.05.

## 3. Results

### 3.1. Patients

From April 2017 to March 2019, 1777 patients with mCRC were enrolled. Among the 1753 patients included in the study, gene sequencing of the tumor samples was completed for 1613 patients (Figure 1). Of these, 439 patients whose tumor samples were obtained after the initiation of chemotherapy were excluded from the study. To avoid survival bias, 567 patients enrolled more than six months after the initiation of first-line chemotherapy were excluded. Finally, 531 patients (355 with left-sided CRC and 176 with right-sided CC) were included in the prognosis analysis.

The relationships among clinicopathological features, major gene alterations (>5%), and sidedness are shown in Table 1. The proportions of female, *BRAF* V600E and *PIK3CA* variants were significantly higher in patients with right-sided CC than those with left-sided CRC (*p* < 0.01, *p* < 0.01, and *p* = 0.02, respectively). We identified 111 *TP53* GOF variants, including 46 R175H, 15 R248W, 21 R248Q, 0 R249S, 12 R273H, 1 R273L, and 16 R282W (one overlapping R175H and R282W). Six patients had truncating *NOTCH3* among the 41 *NOTCH3* sole variants, which were all found in left-sided CRC. There were no significant differences in *FLT3* and *MYC* copy numbers between left-sided CRC and right-sided CC [median copy number: 6.64 (range: 5.12–32.2) vs. 6.48 (range: 5.12–29.0), *p* = 0.49, and 5.72 (range: 4.87–37.4) vs. 6.22 (range: 4.75–87.8), *p* = 0.54, respectively].

### 3.2. Prognostic Significance of Gene Alterations by Sidedness

The median follow-up time was 25.5 months (range: 0.7–59.7 months), and 349 patients (65.7%) died. The median OS durations were 28.5 and 26.3 months in the left-sided CRC and right-sided CC, respectively (HR 1.17, 95% CI 0.94 to 1.46, *p* = 0.16). Significant differences were noted when comparing OS between left-sided CRC and right-sided CC (Table 2 and Figure 2) in the *TP53* non-GOF variant (median OS: 31.1 vs. 22.3 months, *p* = 0.02) (Figure 2a), *KRAS* wild-type (median OS: 30.7 vs. 25.3 months, *p* = 0.04) (Figure 2b), *NOTCH1* wild-type (median OS: 29.5 vs. 25.2 months, *p* = 0.04) (Figure 2c), *NOTCH1* covariant (median OS: 16.8 vs. 31.2 months, *p* = 0.03) (Figure 2c), *NOTCH3* sole variant (median OS: not reached vs. 26.5 months, *p* = 0.01) (Figure 2d), and *MYC* amplification (median OS: 23.6 vs. 11.7 months, *p* = 0.01) (Figure 2e). The difference in OS between left-sided CRC and right-sided CC was observed in the *TP53* non-GOF variant with respect to the *TP53* variant (Figure 2a), and in the *NOTCH3* sole variant with respect to the *NOTCH3* variant (Figure 2d). There was no significant difference in OS between the *TP53* R273H and R175H variants (median OS: 22.5 vs. 17.4 months, *p* = 0.84). Six patients with truncated *NOTCH3* survived for 28.5–56.8 months, and four of the six patients were alive (Figure 2d).

The multivariate analysis results are shown in Figure 3. The *BRAF* non-V600E and *NOTCH1*, *NOTCH2*, and *NOTCH3* covariants were excluded from the analysis because of their low numbers (Table 1). The *TP53* GOF and *KRAS* variants were independent prognostic factors for poorer prognoses (HR 1.51, 95% CI 1.04 to 2.20, *p* = 0.03, and HR 1.48, 95% CI 1.11 to 1.96, *p* < 0.01, respectively), and the *NOTCH3* sole variant for better prognoses (HR 0.30, 95% CI 0.15 to 0.62, *p* < 0.01), in left-sided CRC (Figure 3a). The *TP53* non-GOF variant, *BRAF* V600E, and *MYC* amplification were independent prognostic factors for poorer prognoses (HR 1.56, 95% CI 1.01 to 2.40, *p* < 0.05, HR 2.56, 95% CI 1.41 to 4.64, *p* < 0.01, and HR 2.29, 95% CI 1.17 to 4.51, *p* = 0.02, respectively) in right-sided CC (Figure 3b).

Regarding the analyses of first-line biologics combined therapy, the *KRAS* variant and *NOTCH3* sole variant remained significant prognostic factors in 225 left-sided CRC patients treated with anti-vascular endothelial growth factor antibody combined therapy (HR 1.58, 95% CI 1.10 to 2.28, *p* = 0.01, and HR 0.26, 95% CI 0.11 to 0.60, *p* < 0.01, respectively), whereas no gene alterations were prognostic in 82 left-sided CRC patients treated with anti-EGFR antibody combined therapy (Table 3). The same analysis could not be performed for right-sided CC because the number of patients was too small to be divided into subgroups.

When *NRAS* variants (*n* = 17, 3.2%) were integrated into the analyses as *RAS* variants combined with *KRAS* variants, the results of the prognostic variants did not change (Appendix A).

## 4. Discussion

We demonstrated that the *TP53* GOF and *KRAS* variants in patients with left-sided CRC, and the *TP53* non-GOF variant, *BRAF* V600E, and *MYC* amplification in patients with right-sided CC, were independent predictors of a shorter OS. Notably, the *NOTCH3* sole variant in patients with left-sided CRC was the only independent predictor of a longer OS. The results are summarized in Table 3. These findings suggest that the broad companion diagnostics used in clinical practice can be useful in predicting the prognosis of both left-sided CRC and right-sided CC.

The distribution of clinicopathological features and gene alterations in left-sided CRC and right-sided CC was similar to that in previous reports, such as the Cancer Genome Atlas Project of human CRC, suggesting that our cohort represents common mCRC [15,20,21].

The OS of the six groups divided by *TP53* status and sidedness was distinctly dichotomized into different prognostic populations: poor survival of *TP53* GOF variants in both sidedness and *TP53* non-GOF variants in right-sided CC (Figure 2a). From each analysis of *TP53* variants, OS significantly differed between left-sided CRC and right-sided CC in the *TP53* non-GOF variant, but not in the *TP53* GOF variant. This was consistent with the results of the previous report, and was considered to be confirmed in our patient cohort [15]. The novel discovery was that the *TP53* GOF and non-GOF variants were independent poor prognostic factors in left-sided CRC and right-sided CC, respectively. The mechanism underlying the different survival behaviors of the *TP53* GOF and non-GOF variants in each side remains unclear. Because the right-sided colon arises from the midgut, and the left-sided colon and rectum arise from the hindgut, the difference in anatomical origin and gut microbiome are the likely reasons, as was recently suggested [22,23,24]. The dichotomous role of *TP53* variants is influenced by the microbial environment in mouse models. In the proximal gut, lower microbial load and diversity are associated with lower levels of gallic acid, which is a polyphenol metabolite produced by gut commensal microbes. In the distal gut, higher microbial load and diversity are associated with higher levels of gallic acid. When the *TP53* GOF variant is exposed to gallic acid, it loses its tumor-suppressive effects, switches to oncogenic effects by activating the WNT signaling pathway, and enhances the proliferation and invasion of tumor cells. Therefore, the influence of the gut microbiome on *TP53* variants should be further investigated.

Distinct *TP53* GOF variants may contribute differently to tumor progression. CRC patients with the *TP53* R273H variant had more progressive disease and poorer survival than those with *TP53* R175H variant [25]. However, we did not find this difference. This analysis, which used the Memorial Sloan Kettering Cancer Center CRC dataset from the Cancer Genome Atlas, included several patients with early-stage CRC. Evaluating the prognostic significance was problematic when limited to patients with mCRC due to the reduced number of patients and events. This issue remains unresolved.

The *KRAS* variant in left-sided CRC, *BRAF* V600E, and *MYC* amplification in right-sided CC were also independent poor prognostic factors. Previous studies have shown that these gene alterations are poor prognostic factors for mCRC [21,26,27,28]. However, the prognostic effect of sidedness remains unclear. These effects can be influenced by subsequent biological drug therapy. The benefit of the anti-EGFR antibody for *RAS* wild-type is more promising in left-sided CRC than right-sided CC [9]. Similarly, the benefit of the BRAF inhibitor combined with the anti-EGFR antibody for the *BRAF* V600E variant is more promising in left-sided CRC than in right-sided CC [29]. Interactions with other factors besides these gene alterations may occur. Although the number of patients with *MYC* amplification in right-sided CC was small, its negative impact on OS was substantial, with a median OS of approximately 12 months (Table 2 and Figure 2e). *MYC* amplification in right-sided CC should be the focus of future research.

Notch3 activates cancer stem cells; promotes tumor growth, invasion, metastasis, angiogenesis, and epithelial–mesenchymal transition; and contributes to chemoresistance. Its overexpression is a poor prognostic factor in patients with mCRC [30,31,32,33,34]. In the present study, we demonstrated that the *NOTCH3* sole variant was an independent favorable prognostic factor in patients with left-sided CRC. Truncating *NOTCH3* among *NOTCH3* sole variants was only observed in left-sided CRC. Truncating *NOTCH3* causes it to lose its intracellular domain, which acts as a transcription factor in the Notch3 receptor signaling pathway. This may result in a favorable prognosis in patients with left-sided CRC who harbor the *NOTCH3* sole variant. We could not confirm this plausible explanation because we did not examine the protein expression of the Notch3 intracellular domain in the same samples.

Our study had several limitations. We did not establish a validation cohort for this study. We investigated the publicly available cBioPortal database targeted sequencing of 471 unresectable colorectal adenocarcinoma samples in MSK-IMPACT™ [21]. However, the multivariate analysis results using the same gene variants as our study were not consistent with our findings, possibly due to differences in measurement systems and target patients and the lack of gene amplification data in the MSK-IMPACT™ cohort. As mentioned above, an immunohistochemical investigation of Notch3 protein expression was necessary. Considering that the primary aim of SCRUM-Japan GI-SCREEN was to enroll patients with advanced gastrointestinal cancers in suitable clinical trials based on their individual genomic profiles, we did not collect information about other clinical details, including later lines of treatment. Patients enrolled within six months of the initiation of first-line chemotherapy were included in this study. These limitations may have introduced survival biases. However, most patients meeting the eligibility criteria can be expected to survive for over six months, and the latter bias seems small. Analyses of the distinct biologics-combined therapy could not sufficiently be performed due to the reduced number of patients, especially in *RAS* wild-type patients treated with the anti-EGFR antibody.

## 5. Conclusions

The prognostic significance of gene alterations differed between left-sided CRC and right-sided CC. This study suggests that *TP53* variant classification into GOF and non-GOF, other than *KRAS*, *BRAF*, and *MYC*, is a useful prognostic biomarker for each-sided mCRC. The *NOTCH3* sole variant in left-sided CRC was found to be the only favorable prognostic factor. These gene alterations are potential stratification factors for clinical trials involving patients with mCRC.

## Figures and Tables

**Figure 1 cancers-15-05172-f001:**
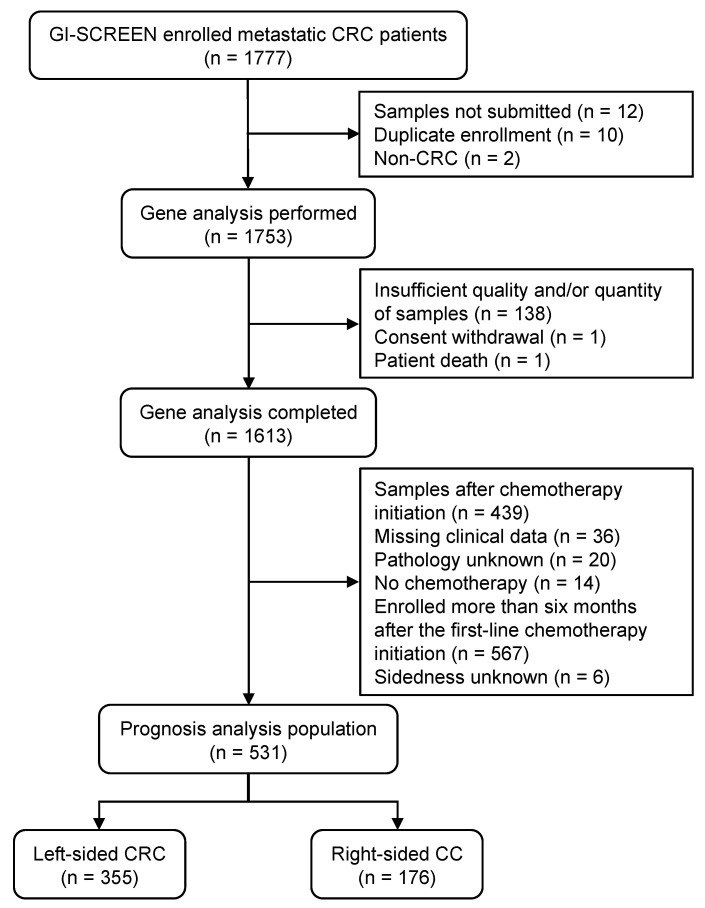
Patient selection flow diagram. CC, colon cancer; CRC, colorectal cancer.

**Figure 2 cancers-15-05172-f002:**
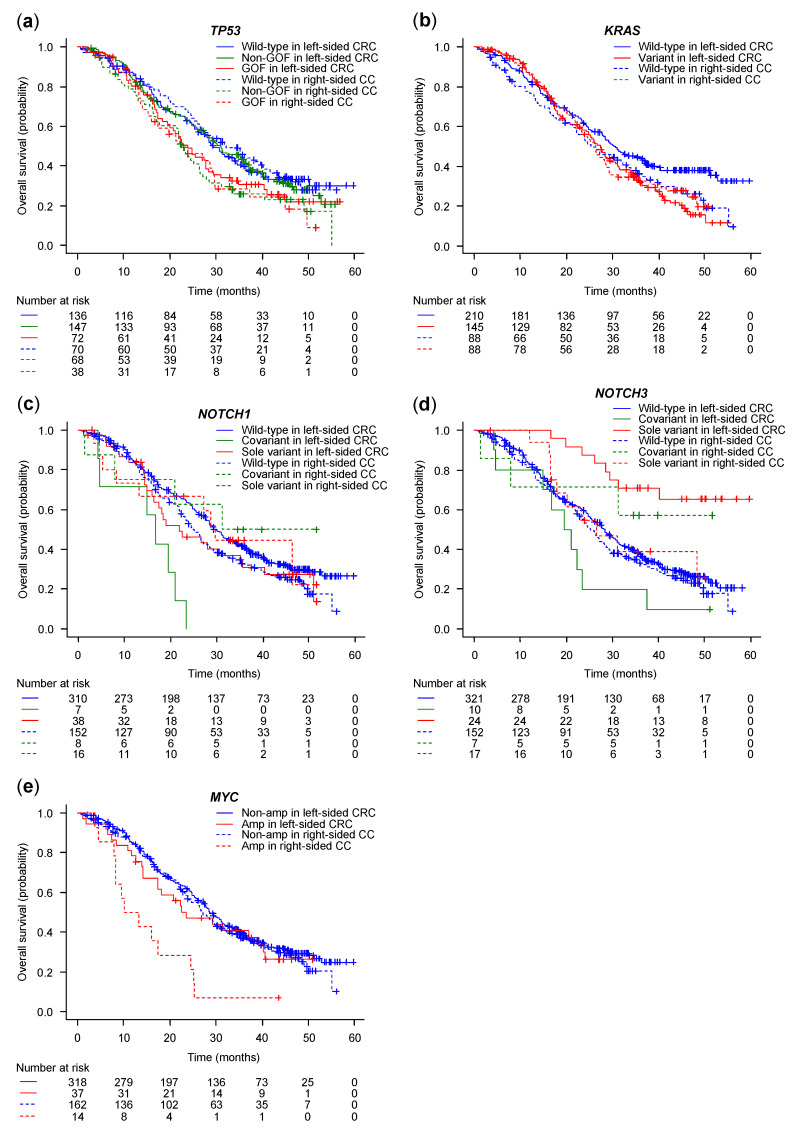
Kaplan–Meier plots of overall survival by gene alterations. (**a**) *TP53*, (**b**) *KRAS*, (**c**) *NOTCH1*, (**d**) *NOTCH3*, and (**e**) *MYC*. Amp, amplification; CC, colon cancer; CRC, colorectal cancer; GOF, gain-of-function; Non-amp, non-amplification; Non-GOF, non-gain-of-function.

**Figure 3 cancers-15-05172-f003:**
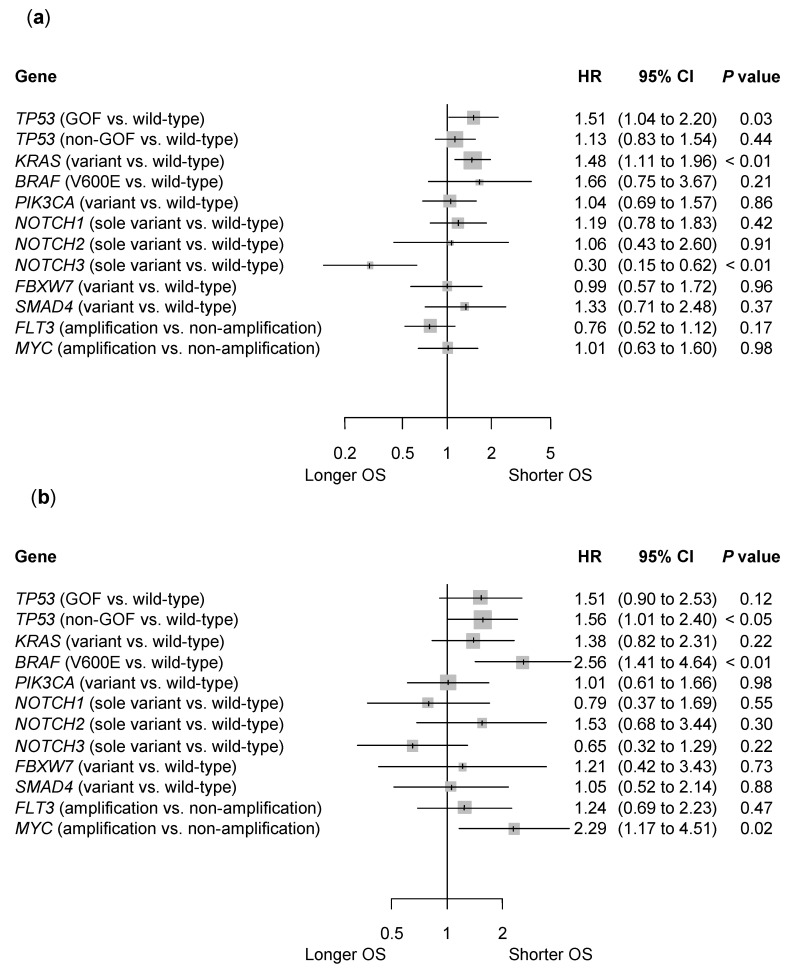
Multivariate analysis. (**a**) Left-sided CRC and (**b**) right-sided CC. CC, colon cancer; CI, confidence interval; CRC, colorectal cancer; GOF, gain-of-function; HR, hazard ratio; Non-GOF, non-gain-of-function; OS, overall survival.

**Table 1 cancers-15-05172-t001:** Patient characteristics, including genes detected at frequencies of more than 5%.

Factor	Group	Left-Sided CRC (*n* = 355) N (%)		Right-Sided CC (*n* = 176) N (%)		*p* Value ^1^
Age	Median, years (range)	63	(25–86)	64	(23–88)	0.28
	<65 y	195	(54.9)	89	(50.6)	0.36
	≥65 y	160	(45.1)	87	(49.4)	
Sex	Male	224	(63.1)	81	(46.0)	<0.01
	Female	131	(36.9)	95	(54.0)	
Histology	Tubular adenocarcinoma	315	(88.7)	145	(83.3)	0.10
	Non-tubular adenocarcinoma ^2^	40	(11.3)	29	(16.7)	
	Adenocarcinoma of unknown differentiation	0		2		
First-line chemotherapy	Anti-VEGF antibody combined therapy	225	(63.4)	137	(77.8)	<0.01
	Anti-EGFR antibody combined therapy	82	(23.1)	13	(7.4)	
	No biological therapy	48	(13.5)	26	(14.8)	
*TP53*	Wild-type	136	(38.3)	70	(39.8)	0.82
	Non-GOF variant	147	(41.4)	68	(38.6)	
	GOF variant	72	(20.3)	38	(21.6)	
*KRAS*	Wild-type	210	(59.2)	88	(50.0)	0.05
	Variant	145	(40.8)	88	(50.0)	
*BRAF*	Wild-type	338	(95.2)	134	(76.1)	<0.01
	Non-V600E variant	4	(1.1)	2	(1.1)	
	V600E variant	13	(3.7)	40	(22.7)	
*PIK3CA*	Wild-type	311	(87.6)	140	(79.5)	0.02
	Variant	44	(12.4)	36	(20.5)	
*NOTCH1*	Wild-type	310	(87.3)	152	(86.4)	0.22
	Covariant	7	(2.0)	8	(4.5)	
	Sole variant	38	(10.7)	16	(9.1)	
*NOTCH2*	Wild-type	341	(96.1)	164	(93.2)	0.20
	Covariant	5	(1.4)	2	(1.1)	
	Sole variant	9	(2.5)	10	(5.7)	
*NOTCH3*	Wild-type	321	(90.4)	152	(86.4)	0.34
	Covariant	10	(2.8)	7	(4.0)	
	Sole variant	24	(6.8)	17	(9.7)	
*FBXW7*	Wild-type	331	(93.2)	167	(94.9)	0.57
	Variant	24	(6.8)	9	(5.1)	
*SMAD4*	Wild-type	338	(95.2)	162	(92.0)	0.17
	Variant	17	(4.8)	14	(8.0)	
*FLT3*	Non-amplification	306	(86.2)	155	(88.1)	0.59
	Amplification	49	(13.8)	21	(11.9)	
*MYC*	Non-amplification	318	(89.6)	162	(92.0)	0.44
	Amplification	37	(10.4)	14	(8.0)	

^1^ *p* values were calculated using the Fisher’s exact test, except for age (range), which was calculated using the Mann–Whitney U test. ^2^ Poorly differentiated, mucinous, and signet ring cell adenocarcinomas. CC, colon cancer; CRC, colorectal cancer; EGFR, epidermal growth factor receptor; GOF, gain-of-function; Non-GOF, non-gain-of-function; VEGF, vascular endothelial growth factor.

**Table 2 cancers-15-05172-t002:** Overall survival analyses.

		Median OS ^1^, Months (95% CI)		
Gene	Group	Left-Sided CRC (*n* = 355)	Right-Sided CC (*n* = 176)	*p* Value ^2^
*TP53*	Wild-type	29.5 (25.3 to 34.6)	32.1 (25.3 to 41.7)	0.79
	Non-GOF variant	31.1 (25.9 to 36.4)	22.3 (17.2 to 26.4)	0.02
	GOF variant	23.9 (18.4 to 27.8)	22.6 (14.0 to 29.7)	0.59
*KRAS*	Wild-type	30.7 (26.6 to 36.4)	25.3 (21.0 to 33.0)	0.04
	Variant	27.4 (23.5 to 30.6)	26.3 (22.1 to 28.7)	0.67
*BRAF*	Wild-type	29.1 (26.6 to 31.4)	27.1 (24.1 to 32.1)	0.57
	Non-V600E variant	26.1 (6.7 to NA)	18.8 (4.6 to NA)	0.32
	V600E variant	14.1 (6.9 to NA)	21.0 (13.7 to 24.5)	0.80
*PIK3CA*	Wild-type	28.5 (26.6 to 31.4)	25.3 (21.9 to 28.7)	0.10
	Variant	25.9 (20.8 to 34.6)	26.7 (23.2 to 38.1)	0.57
*NOTCH1*	Wild-type	29.5 (27.7 to 33.0)	25.2 (22.2 to 28.2)	0.04
	Covariant	16.8 (4.3 to 21.1)	31.2 (1.4 to NA)	0.03
	Sole variant	22.0 (16.1 to 35.2)	29.7 (5.2 to NA)	0.63
*NOTCH2*	Wild-type	28.5 (26.2 to 31.4)	26.3 (22.3 to 29.4)	0.18
	Covariant	22.4 (4.7 to NA)	21.0 (21.0 to NA)	0.77
	Sole variant	19.9 (5.6 to NA)	24.1 (8.0 to 35.2)	0.71
*NOTCH3*	Wild-type	28.4 (25.3 to 31.1)	25.3 (22.2 to 28.6)	0.22
	Covariant	20.3 (4.3 to 23.5)	Not reached (1.4 to NA)	0.10
	Sole variant	Not reached (31.3 to NA)	26.5 (16.5 to NA)	0.01
*FBXW7*	Wild-type	28.8 (25.9 to 31.4)	26.3 (23.2 to 28.7)	0.16
	Variant	28.5 (17.4 to 40.4)	27.0 (9.5 to NA)	0.95
*SMAD4*	Wild-type	29.1 (26.1 to 31.8)	26.3 (23.2 to 29.4)	0.18
	Variant	26.6 (9.7 to 31.2)	26.7 (8.7 to 43.3)	0.99
*FLT3*	Non-amplification	28.4 (24.8 to 31.2)	26.5 (22.6 to 31.2)	0.35
	Amplification	33.9 (26.7 to 40.6)	26.1 (17.2 to 28.6)	0.08
*MYC*	Non-amplification	28.8 (26.6 to 31.6)	27.2 (23.9 to 31.2)	0.37
	Amplification	23.6 (14.2 to 39.4)	11.7 (7.9 to 24.5)	0.01

^1^ Kaplan–Meier estimates of median overall survival; ^2^ *p* values were calculated using log-rank tests. CC, colon cancer; CI, confidence interval; CRC, colorectal cancer; GOF, gain-of-function; NA, not available; Non-GOF, non-gain-of-function; OS, overall survival.

**Table 3 cancers-15-05172-t003:** Results summary of significant prognostic genes by sidedness.

Gene	Group	Frequency Left vs. Right	OS Left vs. Right	Prognosis of Left-Sided CRC			Prognosis of Right-Sided CC
				All (*n* = 355)	Anti-VEGF Antibody Combined (*n* = 225)	Anti-EGFR Antibody Combined (*n* = 82)	All (*n* = 176)
*TP53*	GOF variant	NS	NS	Poor	NS	NS	NS
	Non-GOF variant	NS	Left > Right	NS	NS	NS	Poor
*KRAS*	Variant	NS	NS	Poor	Poor	— ^1^	NS
*BRAF*	V600E variant	Left < Right	NS	NS	NS	— ^1^	Poor
*NOTCH3*	Sole variant	NS	Left > Right	Good	Good	— ^1^	NS
*MYC*	Amplification	NS	Left > Right	NS	NS	NS	Poor

^1^ Not assessed due to the small number of samples (fewer than 5). CC, colon cancer; CRC, colorectal cancer; EGFR, epidermal growth factor receptor; GOF, gain-of-function; Non-GOF, non-gain-of-function; NS, not significant; OS, overall survival; VEGF, vascular endothelial growth factor.

## Data Availability

The authors declare that all variant data used in the conduct of the analyses are available within the article and its Appendix A. To protect the privacy and confidentiality of patients in this study, clinical data are not publicly available in a repository or the Appendix A of the article, but will be made available following reasonable request to the corresponding author.

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
