# Peer review of "Sidedness-Dependent Prognostic Impact of Gene Alterations in Metastatic Colorectal Cancer in the Nationwide Cancer Genome Screening Project in Japan (SCRUM-Japan GI-SCREEN)"

_cancers, 2023, doi:10.3390/cancers15215172_

Round 1

Reviewer 1 Report

Comments and Suggestions for Authors

The manuscript focuses on a retrospective analysis of concomitant molecular laterations in CRC patients according to the role of each molecular subtype represents a technically correct and timely relevant manuscript where few minor integrations shoudl be approached to improve the readibility of the manuscript.

- In the introduction section, please, could the authors overview the clinical landscape of actually approved biomarkers in CRC patients? In my opinion, this point may improve the readibility of the manuscript on this journal

- In the methodological section, please, coudl the authors provide other clinical details about the selcted patient cohort population?

- According to KRAS mutation, please, could the authors show if KRAS mutations play a different role in the clinical stratification of CRC patients?

- In the discussion section, please, could the authors highlight how the implementation of this model in clinical practice may benefit on CRC patients?

Comments on the Quality of English Language

Moderate neglish revision sohuld be approached

Reviewer 2 Report

Comments and Suggestions for Authors

The study is very well described, the manuscript is clearly presented. To my opinion, the main finding of the study is the distinct role of gain-of-function and non-gain-of-function mutations in the TP53 gene.

Suggestions:

Abstract should mention the net numbers of right- and left-sided tumors analyzed.

Introduction should provide sufficiently detailed overview on biology and treatment controversies for left- versus right-side colorectal carcinomas.

Why NRAS mutations were not considered in the analysis?!

The analysis of prognostic impact of KRAS and  BRAF mutations has little meaning and even potentially misleading when no adjustment for actual treatment history is given. MEK inhibitors are not commonly used for the treatment of BRAF-mutated colorectal tumors (lines 282-284). The outcome of the disease in patients carrying BRAF V600E mutation may differ significantly in patients who received or did not BRAF-targeted therapy, and the line of this treatment needs to be taken into account. Even more controversies is related to the treatment of KRAS/NRAS wild-type carcinomas. Some hospitals/doctors consider the side of the tumor while administering panitumumab or cetuximab, while others utilize these drugs for all KRAS/NRAS wild-type patients. Presence or absence of anti-EGFR therapy is a critical factor affecting the outcome of KRAS/NRAS wild-type colorectal cancer.       

Reviewer 3 Report

Comments and Suggestions for Authors

The study is well designed. And the study design, inclusion criteria and molecular genetic analyses are valid and well understood. The article is written in good English and has a good comprehensible structure.

As a small note, it would be useful to list in the cohort of right-sided and left-sided CRC how the distribution úbetween oligometastasized patients compared to advanced metastasized patients is, whether metastase resection was performed in case of oligometastasis and in how many patients in both groups no surgery (primary tumor and/or metastases) was performed. Furthermore, a table with the performed system groups in the compared patient cohorts would be helpful.

With these minor limitations, the work can be recommended for publication without restriction. 

Round 2

Reviewer 2 Report

Comments and Suggestions for Authors

The authors have improved the paper